# Incidence and Risk Factors for Berger’s Space Development after Uneventful Cataract Surgery: Evidence from Swept-Source Optical Coherence Tomography

**DOI:** 10.3390/jcm11133580

**Published:** 2022-06-21

**Authors:** Zhengwei Zhang, Jinhan Yao, Shuimiao Chang, Piotr Kanclerz, Ramin Khoramnia, Minghui Deng, Xiaogang Wang

**Affiliations:** 1Department of Ophthalmology, Wuxi Clinical College, Nantong University, Wuxi 214002, China; weir2008@ntu.edu.cn; 2Department of Ophthalmology, Wuxi No. 2 People’s Hospital, Nanjing Medical University, Wuxi 214002, China; 3Department of Cataract, Shanxi Eye Hospital, Shanxi Medical University, Taiyuan 030002, China; yjh961106@163.com (J.Y.); changsm991101@163.com (S.C.); 4Hygeia Clinic, 80-286 Gdańsk, Poland; p.kanclerz@gumed.edu.pl; 5Helsinki Retina Research Group, University of Helsinki, 00014 Helsinki, Finland; 6The David J. Apple International Laboratory for Ocular Pathology, Department of Ophthalmology, University of Heidelberg, 69120 Heidelberg, Germany; ramin.khoramnia@med.uni-heidelberg.de; 7Department of Cataract, Linfen Yaodu Eye Hospital, Linfen 042000, China; dengminghuimail@163.com

**Keywords:** anterior hyaloid detachment, Berger’s space, phacoemulsification, swept-source optical coherence tomography

## Abstract

Background: This study investigates the incidence and risk factors for the development of Berger’s space (BS) after uneventful phacoemulsification based on swept-source optical coherence tomography (SS-OCT). Methods: Cataractous eyes captured using qualified SS-OCT images before and after uneventful phacoemulsification cataract surgery were included. Six high-resolution cross-sectional anterior segment SS-OCT images at 30° intervals were used for BS data measurements. BS width was measured at three points on each scanned meridian line: the central point line aligned with the cornea vertex and two point lines at the pupil’s margins. Results: A total of 223 eyes that underwent uneventful cataract surgery were evaluated. Preoperatively, only two eyes (2/223, 0.9%) were observed to have consistent BS in all six scanning directions. BS was observed postoperatively in 44 eyes (44/223, 19.7%). A total of 13 eyes (13/223, 5.8%) with insufficient image quality, pupil dilation, or lack of preoperative image data were excluded from the study. A total of 31 postoperative eyes with BS and 31 matched eyes without BS were included in the final data analysis. The smallest postoperative BS width was in the upper quadrant of the vertical meridian line (90°), with a mean value of 280 μm. The largest BS width was observed in the opposite area of the main clear corneal incision, with a mean value >500 μm. Conclusions: Uneven-width BS is observable after uneventful phacoemulsification. Locations with a much wider BS (indirect manifestation of Wieger zonular detachment) are predominantly located in the opposite direction to the main corneal incisions.

## 1. Introduction

Berger’s space (BS), also termed the vitreolenticular interface, hyaloid–capsular interspace, or patellar fossa, is a space located between the posterior lens capsule and anterior hyaloid of the vitreous. These structures attach in a circular manner via thickened hyalo-capsular zonules of the Wieger ligament, the outer limit of which is defined by Egger’s line. Growing evidence suggests that BS is a real and clinically significant space in pathological conditions that can be detected using slit-lamp biomicroscopy [1,2]. Further, Weidle [3] identified the presence of BS in infants by filling the space with an ophthalmic viscosurgical device (OVD) after a small posterior capsulotomy during congenital cataract surgery.

In terms of clinical application, to sever attachments and create a wide interspace, Menapace [4] attempted transzonular capsulo-hyaloidal hydroseparation by rinsing the zonular fibers with fluid or additional triamcinolone acetonide (TA) to initiate or complete anterior hyaloid detachment (AHD). This procedure may improve the patency and visibility of BS to augment the control and feasibility of primary posterior laser capsulotomy (PPLC) in femtosecond laser-assisted cataract surgery (FLACS) with an intact anterior hyaloid membrane that acts as a major barrier between the anterior and posterior segments of the eye. In this regard, adequate imaging of the posterior lens capsule and anterior hyaloid membrane is a prerequisite for safe and effective PPLC. Similarly, viscodissection of BS is also greatly important for manual primary posterior capsulorhexis during the surgery of pediatric cataracts, to avoid vitreous prolapse and destabilization of the intraocular lens [5]. 

Recently, the importance of BS as an anatomical structure for microsurgery has increased. Partial AHD and BS opening are sometimes intentionally induced using hydrostatic pressure to safely and completely remove anterior vitreous hemorrhage [6]. BS can be used as a surgical plane for viscodelamination of the interface between the posterior lens capsule and anterior hyaloid membrane. Lyu et al. [7] reported that retrolental plaques in eyes with pediatric tractional vitreoretinopathy were successfully separated from the posterior lens capsule by blunt tension of cohesive viscoelastic injection into BS with an intact posterior capsule during lens-sparing vitrectomy, because these plaques attach to the anterior hyaloid membrane before invading the BS and posterior lens capsule. Moreover, Kam et al. [8] reported a canal of Petit pneumodissection technique via endoscopy-guided dissection of anatomical planes using filtered air to enable safe and complete separation of the anterior hyaloid of the vitreous from the posterior lens capsule in phakic or pseudophakic eyes. Therefore, both the cataract and vitreoretinal specialist should pay attention to and use this interesting finding to assist or notice the importance of BS or AHD in clinic.

Phacoemulsification is a widely used and safe procedure for the treatment of cataracts. However, rare accidents, such as acute aqueous misdirection syndrome and Descemet membrane detachment, may occur during surgery. In a previous published case of traumatic cataract surgery involving a 41-year-old man with moderate myopia and angle recession in his left eye [9], a large air bubble running into Berger’s space (BS) was noticed during the cortex removal procedure. It is, therefore, of importance to cataract surgeons that BS is a noteworthy anatomical structure and has a possible influence on phacoemulsification.

More recently, BS has been visualized intraoperatively during cataract surgery via real-time intraoperative optical coherence tomography (iOCT) attached to femtosecond laser cataract systems [10,11], or with an operating microscope [12]. Detecting BS before and after cataract surgery using advanced noninvasive optical technology is crucial, as these eyes could be at risk of aqueous misdirection [13]. Studies using spectral-domain optical coherence tomography (SD-OCT) with an anterior segment module have identified BS in pseudophakic patients due to the thinner interface of the intraocular lens compared to that of the natural lens [1]. However, it is insufficient to capture images of BS with SD-OCT in most patients with their natural lenses because SD-OCT imaging fails to reach the depth necessary to visualize the interface between the posterior lens capsule and anterior hyaloid.

Compared to SD-OCT, swept-source optical coherence tomography (SS-OCT) employs a longer wavelength and has a greater scanning depth. Accordingly, it is a more appropriate method to evaluate the anterior hyaloid interface [14,15,16,17]. Currently, the number of SS-OCT devices commercially available in clinical settings is increasing, which may improve our understanding of the physiology and pathology of BS.

The growing number of published articles has signified increasing interest in this anatomical space in recent years. The main objective of this study was to evaluate the influence of phacoemulsification surgery on the change in the postoperative structure of the hyaloid–capsular interspace using a commercially available SS-OCT device with anterior imaging in a relatively large sample, to further clarify the incidence of pre- and post-BS and potential risk factors for the change in postoperative BS.

## 2. Materials and Methods

This retrospective observational study included patients who were willing to undergo cataract surgery at Shanxi Eye Hospital between November 2020 and December 2021. The study was registered online on the International Standard Randomized Controlled Trials website (http://www.controlled-trials.com; accessed on 8 November 2021) with the registration number ISRCTN13860301. All participants provided written informed consent for participation in the clinical examination program and to undergo cataract surgery. This study was conducted in accordance with the tenets of the Declaration of Helsinki. The Institutional Review Board of Shanxi Eye Hospital affiliated with Shanxi Medical University approved the protocol (No. 2019LL130).

The medical records of patients with cataracts who consented to surgery were reviewed. Patients undergoing phacoemulsification cataract surgery were enrolled. Inclusion criteria were as follows: diagnosis of cataracts prepared for surgery, dilated pupil size of 7 mm or larger, no pathological alteration in the anterior segment (such as keratoconus, pseudoexfoliation syndrome, or corneal opacity), no retinal diseases impairing visual function, no previous anterior or posterior segment surgery, and no intraoperative or postoperative complications. Multiple parameters were extracted to determine BS and analyze the related factors. All included patients underwent conventional phacoemulsification. Before and after cataract surgery, each patient underwent complete ocular examination, including best-corrected visual acuity (BCVA), non-contact tonometry, slit-lamp examination, and indirect ophthalmoscopy. Axial length was measured using IOLMaster 700 (Carl Zeiss Meditec, Dublin, CA, USA). All included patients underwent anterior segment SS-OCT using the ANTERION device (software version 1.3.4.0; Heidelberg Engineering, Heidelberg, Germany) with a 1300 nm light source [18,19].

SS-OCT anterior segment imaging pre- and post-surgery was performed by the same technician in a semi-dark room without pupillary dilation, with the patient in the seated position, as reported in a previous study [18]. The ANTERION Metrics App in conjunction with a high-resolution SS-OCT imaging device provides 6 high-resolution (axial resolution < 10 μm and lateral resolution < 30 μm) cross-sectional anterior segment images at 30° intervals (0–180°, 30–210°, 60–240°, 90–270°, 120–300°, and 150–330°) centered on the corneal vertex (Figure 1). The quality of each measurement was examined by an expert, and images that revealed BS were used for further analysis. BS width was manually measured at three points on each scanned meridian line: the central point and two point lines aligned with the corneal vertex and pupil margins, respectively (Figure 2). To identify risk factors for the development of BS after surgery, the 31 matched eyes without postoperative BS were selected for comparison analysis.

## 3. Surgical Procedure

All phacoemulsification surgeries were performed under local anesthesia by a single surgeon (X.G.W.). Phacoemulsification parameters were set as follows: continuous linear mode was used with ultrasound (US) power up to 40%, vacuum was linear to 500 mmHg, and bottle height (recorded as irrigation pressure), to provide passive infusion, was set at 95–110 cm above eye level. A 2.2 mm clear corneal incision was made superior-temporally in the right eye or superior-nasally in the left eye (120°). After the creation of a continuous curvilinear capsulorhexis, phacoemulsification (horizontal phaco-chop technique; angle 30° phaco tip No. DP8730 with outer diameter of 0.9 mm and inner diameter of 0.70–0.5 mm) was followed by aspiration of the cortical remnants using the Stellaris^®^ MICS^TM^ system (Bausch + Lomb, Rochester, NY, USA). A foldable monofocal hydrophobic acrylic intraocular lens (IOL HOYA PY60AD) was implanted into the capsular bag. In all patients, no sutures were used to close the incisions.

## 4. Statistical Analyses

Statistical analyses were performed using SPSS software (version 21.0, SPSS, Chicago, IL, USA). The normality of the distribution of continuous variables was assessed using a one-sample Kolmogorov–Smirnov test prior to significance testing. Normally distributed data were analyzed with an independent Student’s *t*-test. Non-normally distributed data were analyzed using the Mann–Whitney U-test. Correlations were assessed using Pearson’s correlation analysis. All normally distributed values of continuous variables are expressed as the mean ± standard deviation (SD), and non-normally distributed values are expressed as median and interquartile ranges (IQR). All tests were two-tailed with a significance level of 5%.

## 5. Results

All 223 eyes that underwent uneventful cataract surgery have undergone a preliminary analysis. In 44 eyes (19.7%), BS could be visualized using an SS-OCT device postoperatively; however, 13 eyes were excluded from the study, as the OCT images demonstrated low quality (4 eyes), had pupil dilation (5 eyes), or had no preoperative data (4 eyes). Ultimately, 31 eyes of 30 patients (21 eyes of 20 patients with age-related cataracts, 3 eyes of 3 patients with metabolic cataracts, 3 eyes of 3 patients with complicated cataracts, 2 eyes of 2 patients with glucocorticoid-induced cataracts, 1 eye of 1 case with high myopia, and 1 eye of 1 case with congenital cataracts) with clearly defined BS after uneventful phacoemulsification were included in the study. These results were matched with 31 eyes from 25 patients (16 eyes of 12 patients with age-related cataracts, 9 eyes of 8 patients with metabolic cataracts, 4 eyes of 3 patients with glucocorticoid-induced cataracts, and 2 eyes of 2 patients with high myopia) without preoperative BS to analyze the risk factors for the development of BS. The clinical characteristics of 30 patients with and 25 patients without postoperative BS are presented in Table 1. Preoperative BS was observed in only two eyes (0.9%). No significant differences were observed in age, axial length, lens thickness, and intraocular pressure between eyes with and without BS. BS development was associated with a higher irrigation pressure (*p* < 0.001) and shorter surgery duration (*p* = 0.021).

The mean follow-up time for anterior SS-OCT imaging after cataract surgery was 24.1 days postoperatively. The mean times of OCT imaging for eyes with and without BS were 14.0 days (range, 1–57 days) and 34.2 days (range, 1–395 days) after surgery, respectively. BS persisted for more than 25 days after cataract surgery in 10 of these eyes. The smallest BS width was observed in the upper vertical meridian line (90°), with a mean value of 280 ± 202 μm. The largest BS width was observed in the opposite area of the main clear corneal incision (240°), with a mean value of 557 ± 352 μm (Figure 3 and Figure 4). Hyperreflective postoperative material was noted in BS in six eyes (Figure 5).

Pearson correlation analysis revealed no significant correlation between postoperative BS width at all meridians and ocular parameters (axial length, lens thickness, and intraocular pressure), surgical parameters (surgery time and irrigation pressure), or age (all *p* values > 0.05). 

## 6. Discussion

Evidence of direct communication between the anterior chamber and BS during phacoemulsification cataract surgery has been inconclusive. The current findings support the hypothesis that uneventful cataract surgery may induce AHD and provides evidence of the volume characteristics of the created space. We observed that 44 eyes (19.7%) presented with BS after phacoemulsification. A previous study reported that BS can be clearly visualized in 81% of cataract cases immediately after IOL implantation with femtosecond laser-integrated SD-OCT [10]. More recently, Anisimova et al. [20] successfully identified BS in 21 cases (75%) via intraoperative optical coherence tomography (iOCT) during phacoemulsification, and in 23 cases (82%) with SD-OCT postoperatively. This difference may be due to the different characteristics of the included populations, different equipment employed, and different time intervals for scanning. For instance, the study by Anisimova et al. [20] included many patients with pseudoexfoliation syndrome (PEX) whose ciliary zonule was inherently weakened [21]. Of note, BS persisted for more than 25 days (maximum of 57 days) after cataract surgery in 10 eyes in our study, suggesting that AHD cannot recover once it occurs either partially or completely.

In the two eyes with preoperative BS, the size of BS in all directions was relatively symmetrical and uniform (Figure 1). However, BS lost its symmetry in all directions, and the largest width appeared in the area opposite the main corneal incision (Figure 4). This phenomenon could predominantly be due to the impact force of the irrigation fluid flow, which may cause a local crevice of the Wieger ligament in the opposite area through the ciliary zonule, resulting in local AHD (Figure 6). We speculate that BS width is related to the extent of Wieger ligament damage, whereby greater BS width indicates greater extent of Wieger ligament damage.

Phacoemulsification cataract surgery may cause posterior vitreous detachment (PVD) [22,23]. However, there is growing interest in AHD after uneventful cataract surgery [20]. Based on visualization using TA, BS has been recognized as a sac-like structure with a septum located behind the lens that divides BS into a two-thirds temporal and one-third nasal formation [24]. The advent of advanced imaging technology, particularly SS-OCT, has facilitated in vivo visualization of the anatomical structure of the vitreolenticular interface [25].

Despite its rare involvement in ocular pathology, BS can be visualized in certain pathological conditions, such as ocular trauma [26,27], spontaneous vitreous hemorrhage [1], ocular surgeries [3,28,29], pigment dispersion syndrome [30], and idiopathic conditions [31]. Accordingly, BS is an actual space that may constitute a key site of pathology. Shah et al. [32] reported a preterm neonate with type 1 retinopathy of prematurity who presented with hemorrhage in BS immediately after intravitreal bevacizumab injection. The hemorrhage resolved completely at 18 weeks postoperatively, and the crystalline lens remained clear. Therefore, specific attention should be paid to disease or surgical manipulations potentially related to BS.

BS is too narrow to be identified in the presence of a natural crystalline lens, even with modern diagnostic tools. As such, this space is unobservable in most cataract patients preoperatively because the posterior capsule of the lens adheres throughout its extension to the anterior vitreous. As reported in a previous study using SD-OCT with the anterior pole module (Cirrus Lumera 700 Carl Zeiss Meditec), only 3 out of 90 patients presented with BS [25]. Similarly, we only identified 2 eyes with preoperative BS out of 223 eyes using SS-OCT. However, potential BS may develop into an obvious space with varying widths immediately after lens extraction from the capsular bag, accompanied by forward movement of the posterior lens capsule. This would increase the probability of occurrence of BS during and after cataract surgery.

Crucially, BS may be enlarged by excessive irrigation fluid circulation that moves through weakened zonules and incomplete attachment of the Wieger ligament during phacoemulsification. Vasavada et al. [33] reported that using high fluidic parameters during phacoemulsification caused partial AHD with an intact posterior lens capsule. We observed that eyes with BS had higher irrigation pressures (*p* < 0.001, Table 1). However, higher irrigation per se may cause higher pressure and damage to zonular fibers, which facilitates the entry of irrigation fluid into BS. Therefore, in cataract patients with weakened zonular fibers, such as high myopia and PEX, irrigation pressure should be reduced appropriately to mitigate damage to zonules and decrease AHD incidence. Surgeons should also carefully adjust the ratio between the bottle height and flow rate to achieve a balanced state, which may enhance the stability of fluid circulation and safety during surgery.

The presence or increasing width of BS or AHD may lead to complications during or after cataract surgery. Vasavada et al. [33] reported that the use of high bottle heights and aspiration flow rates may have detrimental consequences on the anterior vitreous face, which are clinically undetectable. This causes decompartmentalization and allows the diffusion of infectious microbes and inflammatory mediators through zonules into BS, consequently increasing the risk of vitritis, macular edema, or even endophthalmitis. However, in this study, no significant correlation was identified between BS width and ocular parameters (axial length, lens thickness, and intraocular pressure), surgical parameters (surgery time and irrigation pressure), or age, although eyes with BS had higher irrigation pressure. This could be due to the small sample size, which may have obscured any statistical correlation. Notably, longer surgical time was observed in the group without postoperative BS. This may be attributed to the recording of surgical time as the whole operation time, rather than just the phaco and irrigation/aspiration time. Ideally, a comprehensive consideration of irrigation pressure and surgical time (phaco and irrigation/aspiration) may better reflect the effects of cataract surgery on the vitreolenticular interface.

AHD is associated with increased instability of the posterior lens capsule (due to loss of Wieger ligament fixation), and is a latent risk factor for posterior capsule rupture during the irrigation/aspiration step of phacoemulsification, thereby increasing the risk of posterior capsule rupture. Partial AHD may also partly contribute to the pathogenesis of acute aqueous misdirection syndrome; however, no cases of BS or AHD after phacoemulsification presenting with acute aqueous misdirection syndrome were noted in our study or previous studies [12,20]. Nevertheless, the complex nature of this syndrome may involve changes in the anatomical structure of the anterior vitreolenticular interface as well as the collection of irrigation fluid or OVD in BS. 

Previous studies have reported the presence of material in BS after uneventful cataract surgery [20,34]. Lenticular zonular insufficiency or partial AHD may provide access to BS from the anterior chamber during irrigation and aspiration, resulting in the entry of medication, residual lens material, and blood cells into this space. Using iOCT, Anisimova et al. [20] reported the presence of BS and penetration of lens fragments into the hyaloid–capsular interspace. In 767 consecutive phaco cases, Kam et al. [34] identified material in BS in 386 eyes (50.3% of cases), the majority of which was putative lens material (46.5% of all cases), with two cases confirmed by histological investigation. In contrast, another study reported that no retrocapsular lens fragments could be identified cytopathologically after uneventful phacoemulsification [35]. Although we could not confirm that the material (uniformly hyper-reflective signal in SS-OCT images) was in BS in the six eyes analyzed herein, we speculate that the material may be OVD or small residual lens fragments migrating from the anterior chamber through the zonular network and detached Wieger ligament (Figure 5).

Due to its retrospective nature, this study had several limitations. First, the limited field of view was a study limitation due to the lack of mydriasis for imaging. Nonetheless, the effective scan captured useful information from the zone within the pupil diameter, as the iris prevented deeper light propagation. Second, we did not evaluate the stability of the anterior chamber depth during phacoemulsification, which is a factor affecting zonular fiber function.

## 7. Conclusions

The Berger’s space can be visualized by SS-OCT not only before but one day or more than one month after cataract surgery. The presence of postoperative BS may be due to the partial AHD during cataract surgery, and the largest width appeared in the opposite area to the main corneal incision, unlike the size of it in all directions being relatively symmetrical and uniform preoperatively.

## Figures and Tables

**Figure 1 jcm-11-03580-f001:**
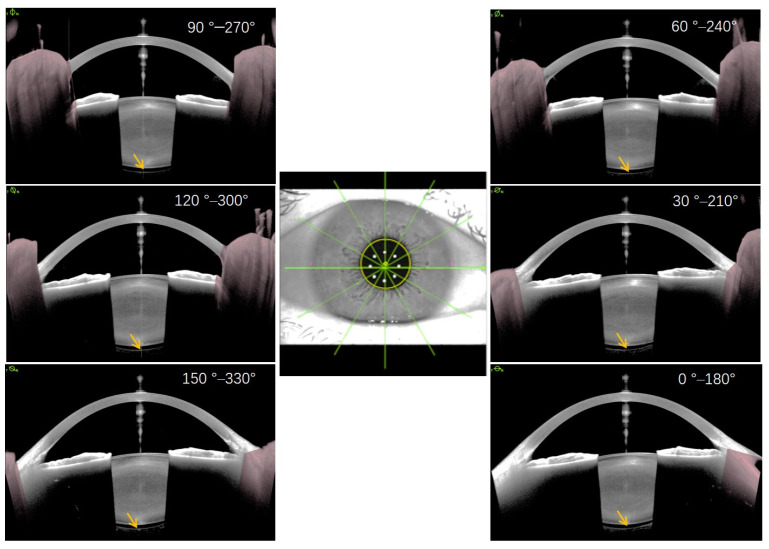
Six high-resolution, cross-sectional anterior segment images with 30° intervals centered on the corneal vertex in a 48-year-old myopic cataract patient with an axial length of 28.70 mm. Yellow arrows indicate Berger’s space with a natural lens. Of note, the size of Berger’s space in all directions is relatively symmetrical and uniform (around 254 μm).

**Figure 2 jcm-11-03580-f002:**
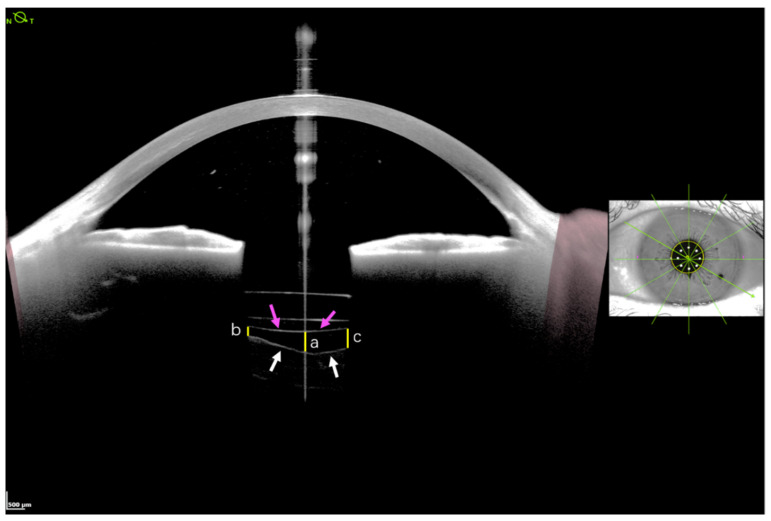
The measurement locations used to determine the width of Berger’s space. The width of Berger’s space, defined as the vertical distance between posterior lens capsule (pink arrows) and anterior vitreous hyaloid (white arrows), was manually measured at three points of each scanned meridian line: the central point line (**a**) and the two point lines (**b**,**c**) at the margins of the pupil (yellow solid lines).

**Figure 3 jcm-11-03580-f003:**
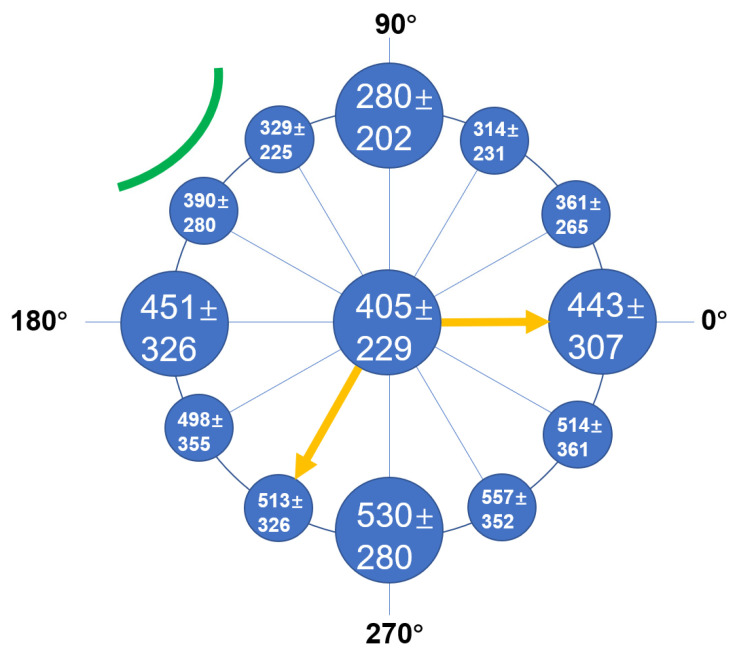
The mean ± standard deviation (SD) width of Berger’s space at three points (as illustrated in Figure 2) at each meridian line. The green curved line indicates the main clear corneal incision for the phacoemulsification tip. The region indicated by the two yellow arrows represents the main impact area of irrigation fluid during cataract surgery. The mean width of Berger’s space opposite to the main clear corneal incision was the largest, which impacted the influence of irrigation fluid circulations.

**Figure 4 jcm-11-03580-f004:**
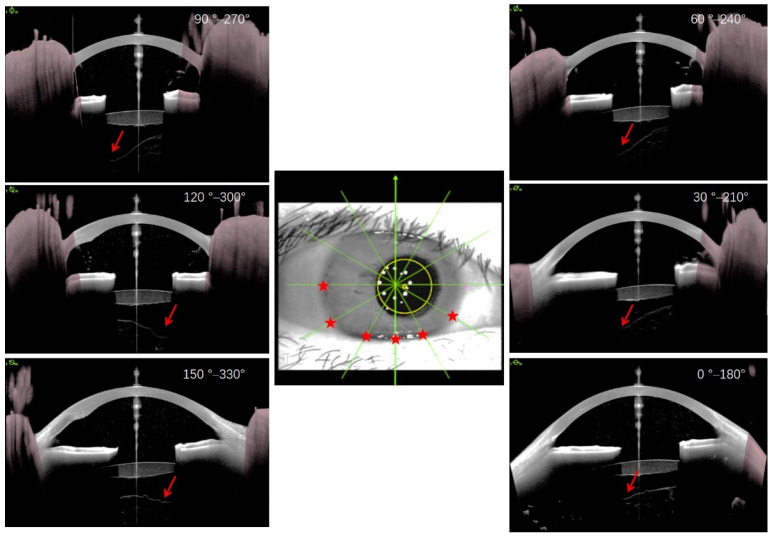
A representative eye postoperatively presented with uneven Berger’s space. Berger’s space lost its symmetry in all directions and the larger width was predominantly observed in the opposite area of the main clear corneal incision (red arrows in cross-sectional optical coherence tomography (OCT) images and red stars in eye plane image).

**Figure 5 jcm-11-03580-f005:**
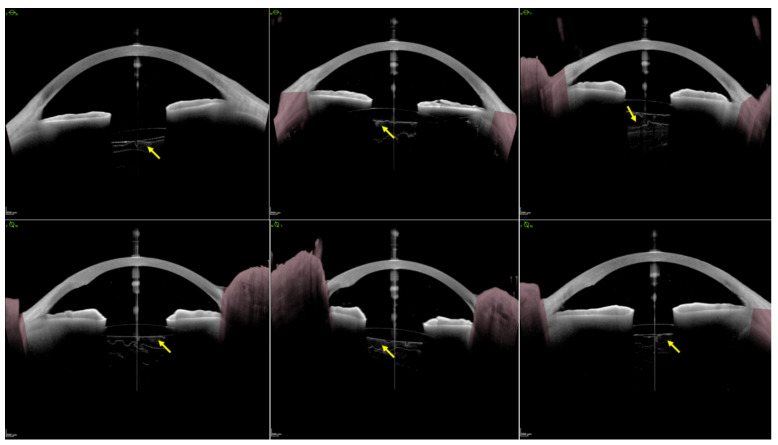
Scattered hyperreflective material (yellow arrows) in Berger’s space was observed in six eyes postoperatively in less than 7 days.

**Figure 6 jcm-11-03580-f006:**
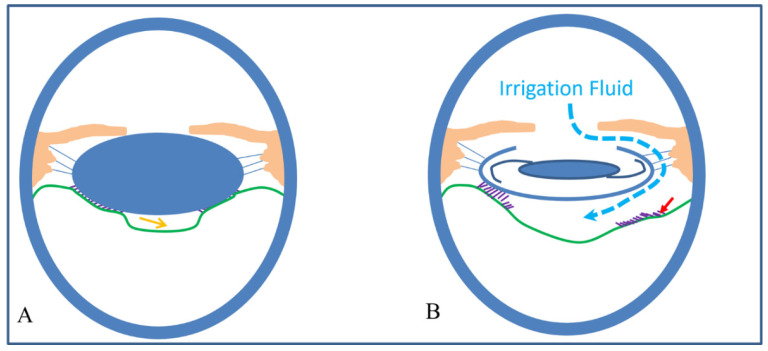
Schematic diagram of Wieger zonular damage and increasing width of Berger’s space during phacoemulsification. (**A**) The yellow arrow indicates Berger’s space with natural lens before surgery (the green line indicated as the anterior hyaloid of the vitreous; the purple lines indicated the Wieger zonular). (**B**) The curved dash line indicates extensive irrigation fluid through the zonular network, resulting in partial Wieger zonular damage (red arrow).

**Table 1 jcm-11-03580-t001:** Clinical characteristics of patients with and without postoperative Berger’s space (BS).

	With BS	Without BS	*p*
N	30	25	
Eyes (OD/OS)	31 (18/13)	31 (18/13)	
Age (years)	66.0 ± 14.2	64.1 ± 14.3	0.635 *
Axial length (mm)	23.22 ± 1.14	23.70 ± 1.91	0.239 *
Lens thickness (mm)	4.13 ± 0.50	4.26 ± 0.56	0.443 *
Intraocular pressure (mmHg)			
Before surgery	15.87 ± 2.50	16.81 ± 2.37	0.136 *
After surgery	17.61 ± 1.61	17.03 ± 1.58	0.157 *
Surgical time (min) ^†^	12.48 ± 3.95	14.58 ± 2.98	0.021 *
Irrigation pressure (cm H_2_O, median (IQR))	110 (110–105)	105 (110–101)	<0.001 ^#^

* Independent Student’s *t*-test, ^#^ Mann–Whitney U test. ^†^ Surgical time was calculated from the beginning of the side incision to the end of the watertight incision closure.

## Data Availability

The datasets used during the current study are available from the corresponding author.

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
