# Peer review of "Incidence and Risk Factors for Berger’s Space Development after Uneventful Cataract Surgery: Evidence from Swept-Source Optical Coherence Tomography"

_jcm, 2022, doi:10.3390/jcm11133580_

Round 1

Reviewer 1 Report

I have read with great interest the manuscript titled: Incidence and risk factors for Berger’s space development after uneventful cataract surgery: evidence from swept-source optical coherence tomography.

Honestly it is a brilliant article, very well documented and super interesting. I have learned a lot by reading it and I would like to thank and congratulate the authors for their work. I think it provides very interesting insight into the anatomy of the lens-vitreous complex, which is so unknown to many ophthalmologists.

Some comments:

I do not understand in results section why author said: ‘’ In 44 eyes (19.7%), BS could be visualized using an SS-OCT device postoperatively’’ and then ‘’Of 223 eyes, 31 eyes of 30 patients (21 eyes of 20 patients with age-related cataracts, 3 eyes of 3 patients with metabolic cataracts, 3 eyes of 3 patients with complicated cataracts, 2 eyes of 2 patients with glucocorticoid-induced cataracts, 1 eye of 1 case with high myopia, and 1 eye of 1 case with congenital cataracts) were observed to have BS after uneventful phacoemulsification. ‘’ Please explain or fix it

Final of discussion and conclusion are similar: please remove the final of the discussion to avoid duplications.

Author Response

  1. I do not understand in results section why author said: “In 44 eyes (19.7%), BS could be visualized using an SS-OCT device postoperatively” and then “Of 223 eyes, 31 eyes of 30 patients (21 eyes of 20 patients with age-related cataracts, 3 eyes of 3 patients with metabolic cataracts, 3 eyes of 3 patients with complicated cataracts, 2 eyes of 2 patients with glucocorticoid-induced cataracts, 1 eye of 1 case with high myopia, and 1 eye of 1 case with congenital cataracts) were observed to have BS after uneventful phacoemulsification. ” Please explain or fix it.

Response- Thank you for your comment. In all 44 eyes, 13 eyes with insufficient image quality, pupil dilation, or lack of preoperative image data were excluded from the study. Therefore, only 31 eyes were included in the final data analysis. We have made a clearer expression in the revised manuscript.

  1. Final of discussion and conclusion are similar: please remove the final of the discussion to avoid duplications.

Response- Thank you for your suggestion, we have removed the final of the discussion in the revised manuscript.

Reviewer 2 Report

This is an interesting study investigating Berger's space after cataract surgery. The authors found an association of the occurence of BS and certain parameters of phaco surgery. No association was found with age, axial length, lens thickness and Intraocular pressure. BS was largest opposite to the main corneal incision.

BS is interesting because it could contribute to post- and intraoperative complications and plays a role for safe posterior capsulorhexis.

Therefore the study adds essential knowledge to an interesting issue. 

Abstract

Please indicate at first how many eyes in total were evaluated. 

The temporal and inferior quadrants are on the opposite of the main clear corneal incision only on the left eye if the surgeon is right-handed. Please clarify.

Introduction

Please describe at first what and where Berger's space actually is. 

Mention the importance of BS for FLACS or congential cataract surgery in the introduction (e.g. Menapace 2008).

Materials and Methods

It may be difficult to get the data retrospectively, but I would suggest to add some more patient data (e.g. posterior vitreous detachment, floppy iris syndrome, usage of pupil expansion rings, PEX, occurence of trampolining during procedure). As BS could play a pivotal role for the development of postoperative complications it would be interesting if patients developed Irvine-Gass syndrome or PCO after cataract surgery.

As BS seems to disappear after some time it is crucial to indicate the time interval between the cataract surgery and the anterior segment imaging.

Please describe the surgical procedure more in detail. How was the irrigation pressure measured. Which method was used (divide and conquer, phaco chop)? How were the 223 cataracts selected? 

Results

How were the 31 eyes without BS for the comparative analysis selected? 

Does table 1 indicate means and standard deviation - please describe. If irrigation pressure is non-normally distributed I would suggest to indicate median and IQR. Describe outliers of irrigation pressure (mature catarcts, PEX?). 

Discussion

Some parts of the discussion describing BS should be moved to the introduction.

The authors could explain why they found BS in significantly less cases than prescribed earlier. Maybe the time interval between surgery and imaging is a good starting point. 

In conclusion the study provides interesting add to our knowledge about BS. I hope my suggestions can help to improve the manuscript.

Author Response

  1. Abstract: Please indicate at first how many eyes in total were evaluated.

Response- We added “A total of 223 eyes that underwent uneventful cataract surgery were evaluated” as a first sentence in the Results.

  1. Abstract: The temporal and inferior quadrants are on the opposite of the main clear corneal incision only on the left eye if the surgeon is right-handed. Please clarify.

Response- Thank you for valuable suggestion. In the present study, all the surgeries were taken by Dr. Xiaogang Wang who is right-handed. Whether in the left eye or the right eye, the largest BS width was observed in the opposite area of the main clear corneal incision (230°-360°). To make the description more proper, we used the angle direction to show the changing area. We corrected this sentence in the Results.

  1. Introduction: Please describe at first what and where Berger's space actually is.

Response- We updated this information in the revised manuscript.  

  1. Introduction: Mention the importance of BS for FLACS or congenital cataract surgery in the introduction (e.g. Menapace 2008).

Response- We have added the importance of BS for FLACS or congenital cataract surgery in the Introduction (Reference #5 in the revised manuscript).

  1. Materials and Methods: It may be difficult to get the data retrospectively, but I would suggest to add some more patient data (e.g. posterior vitreous detachment, floppy iris syndrome, usage of pupil expansion rings, PEX, occurrence of trampolining during procedure). As BS could play a pivotal role for the development of postoperative complications it would be interesting if patients developed Irvine-Gass syndrome or PCO after cataract surgery.

Response- In our retrospective study, no patient was found with floppy iris syndrome, PEX, or usage of pupil expansion rings. During the surgery, the anterior chamber depth remained stable in my mind. No patient was found Irvine-Gass syndrome and PCO until last follow-up.

  1. Materials and Methods: As BS seems to disappear after some time it is crucial to indicate the time interval between the cataract surgery and the anterior segment imaging.

Response- We added time interval information in the Result part. In our study, we found that BS persisted for more than 57 daysafter cataract surgery, suggesting that AHD may not recover once it occurs either partially or completely. We will extend the follow-up time point to observe the BS changing tendency in our further study.

  1. Materials and Methods: Please describe the surgical procedure more in detail. How was the irrigation pressure measured? Which method was used (divide and conquer, phaco chop)?

Response- The irrigation pressure was recorded as bottle height (cm H2O) showed on the phaco machine screen. Horizontal phaco-chop technique was used during phacoemulsification. We have supplemented this part in the revised manuscript.

  1. Materials and Methods: How were the 223 cataracts selected?

Response- The 223 cataracts were selected because they were consecutively scanned by anterior SS-OCT between November 2020 and December 2021 according to inclusion and exclusion criteria in the Materials and Methods.

  1. Results: How were the 31 eyes without BS for the comparative analysis selected?

Response- The matched 31 eyes without BS for the comparative analysis were selected according to the similar age and ocular parameters of eyes with BS. We made a revised statement in the Materials and Methods.

  1. Results: Does table 1 indicate means and standard deviation - please describe. If irrigation pressure is non-normally distributed I would suggest to indicate median and IQR. Describe outliers of irrigation pressure (mature catarcts, PEX?).

Response- We added a sentence “All normally distributed values of continuous variables are expressed as the mean ± standard deviation (SD) and non-normally distributed values are expressed as median and interquartile ranges (IQR)” in the Statistical analyses. Based on individual ocular parameters, the bottle height (irrigation pressure) was set in the range of 95 to 110 cmH2O. We have stated in the Materials and Methods.

  1. Discussion: Some parts of the discussion describing BS should be moved to the introduction.

Response- Thank you for your comment, we made some amendment in the revised manuscript.

  1. Discussion: The authors could explain why they found BS in significantly less cases than described earlier. Maybe the time interval between surgery and imaging is a good starting point.

Response- Indeed, the time interval between surgery and OCT imaging needs to be taken into consideration, because potential BS may develop into an obvious space with varying widths immediately (using intraoperative OCT imaging) after lens extraction from the capsular bag accompanied by forward movement of the posterior lens capsule. We also added this information into the discussion part.

Round 2

Reviewer 2 Report

Thank you for adressing all my comments. I do not have any further suggestions.